# The Sodium/Calcium Exchanger PcNCX1-Mediated Ca²⁺ Efflux Is Involved in Cinnamaldehyde-Induced Cell-Wall Defects of *Phytophthora capsici*



Zhongqiang Qi [1], Lina Li [2], Cunfa Xu [3], Muxing Liu [4], Yousheng Wang [5], Li Zhang [6], Jian Chen [7], Haiyan Lu [7] and Zhiqi Shi [7,*]

[1] Laboratory for Food Quality and Safety, State Key Laboratory Cultivation Base of Ministry of Science and Technology, Institute of Plant Protection, Jiangsu Academy of Agricultural Science, Nanjing 210014, China; 20130019@jaas.ac.cn

[2] Tianmen Green Food Management Office, Jingling Sub-District Office, Tianmen 431700, China; llnhphp1314@163.com

[3] Central Laboratory, Jiangsu Academy of Agricultural Science, Nanjing 210014, China; jaasxucunfa@163.com

[4] Department of Plant Pathology, College of Plant Protection, Nanjing Agricultural University, Nanjing 210095, China; liumuxing@njau.edu.cn

[5] China Tobacco Jiangsu Industrial Co., Ltd., Nanjing 210011, China; hy2005002@jszygs.com

[6] College of Plant Protection, Shandong Agricultural University, Taian 271028, China; lilizhang324@163.com

[7] Laboratory for Food Quality and Safety, State Key Laboratory Cultivation Base of Ministry of Science and Technology, Institute of Food Safety and Nutrition, Jiangsu Academy of Agricultural Sciences, Nanjing 210014, China; chenjian@jaas.ac.cn (J.C.); haiyanlu@jaas.ac.cn (H.L.)

[*] Correspondence: shizhiqi@jaas.ac.cn

**Abstract:** *Phytophthora capsici* is one of the devastating pathogens, causing foliar blight, root rot, and fruit rot in peppers. Cinnamaldehyde (CA) is a natural compound coming from *Cinnamomum cassia*. The medicinal properties of CA have been widely identified. Limited knowledge is known about the application of CA in agriculture. In this study, CA significantly inhibited *P. capsici*, which further suppressed Phytophthora blights in both pepper seedlings and pepper fruits. Treatment with CA resulted in collapsed and fragmented hyphae, accompanying the increase in MDA (malondialdehyde) content and the decrease in intercellular glycerol content in hyphae. CA also inhibited the growth of wild type yeast. The yeast mutant ΔYvc1 with a deletion of Yvc1 (a Ca²⁺ transporter) showed decreased sensitivity to CA. The transformation of *PcNCX1*, a sodium/calcium exchanger from *P. capsici*, into ΔYvc1 restored its sensitivity to CA. The transformant carrying *PcNCX1* also showed restored Ca²⁺ efflux upon CA treatment. RNA-seq analysis showed that CA treatments resulted in the down-regulation of a set of genes encoding for calcium-related proteins. Collectively, our study demonstrates that the antifungal activity of CA against *P. capsici* may be associated with PcNCX1-mediated Ca²⁺ efflux. Our results provide crucial insights into the antimicrobial action of CA.

**Keywords:** calcium; cell wall integrity; cinnamaldehyde; PcNCX1; *Phytophthora capsici*

## 1. Introduction

Phytophthora blights caused by *Phytophthora capsici* are an increasingly severe disease on solanaceous and cucurbitaceous hosts, such as pepper, cucumber, tomato, eggplant, and pumpkin [1]. Chemical fungicides are frequently used to control Phytophthora blight diseases. However, the long-term application of chemical fungicides not only leads to the resistance of *P. capsici* but also poses potential threats on human health and environment [2]. Consequently, environmentally friendly fungicides need to be developed in order to control Phytophthora blight diseases effectively.

Cinnamaldehyde (CA), a major constituent of cinnamon essential oils, has been widely used as food additives and medicine due to its antimicrobial activities and pharmacological effects [3]. CA can modulate physiological changes in host mammalian cells by regulating several intrinsic signals such as $Ca^{2+}$, nitric oxide (NO), and adenosine monophosphate-activated protein kinase (AMPK) [4]. In the Gram-positive bacterium *Staphylococcus aureus*, CA can inhibit its hemolytic activity on human erythrocytes and reduce its adherence to latex [5]. Moreover, In *Streptococcus mutans*, CA decreased biofilm biomass and metabolism at sub-MIC (Minimum inhibitory concentrations) concentrations [6]. In *Aspergillus brasiliensis*, CA exhibits a high in vitro antifungal effect by altering the morphology of the hyphae [7]. It has been reported that CA has significant antifungal activity against *Colletotrichum lagenarium*, an important plant-pathogenic fungus causing the anthracnose of cucumbers [8]. Our previous study suggested that CA had great potential to control Phytophthora blight disease by promoting pepper root growth and inhibiting the growth of *P. capsici* [9]. $Ca^{2+}$ efflux has been closely linked to the antifungal activity of CA against *P. capsici* [9]. However, the molecular mechanism for CA-mediated $Ca^{2+}$ efflux in *P. capsici* remains unclear.

$Ca^{2+}$ is a secondary messenger involved in numerous signaling pathways in eukaryotic cells [10]. In filamentous fungi, $Ca^{2+}$ regulates a diverse array of biological processes including cell cycle, sporulation, hyphal morphogenesis, infection structure differentiation, and pathogenesis [11,12]. The disruption of $Ca^{2+}$ homeostasis results in defects of fungal cell wall integrity, further inhibiting fungal growth [13,14]. $Ca^{2+}$ homeostasis is maintained by $Ca^{2+}$ channels, pumps, and transporters. The $Na^+/Ca^{2+}$ exchanger (NCX) is the major pathway for transport of $Ca^{2+}$ out of the cytoplasm, a putative *NCX* gene, *Arabidopsis* NCX-like (*AtNCL*), contributed to $Ca^{2+}$ homeostasis under abiotic stress [15]. The yeast vacuolar conductance 1 (Yvc1, also named TRPY1) is a $Ca^{2+}$ transporter, which mediates $Ca^{2+}$ release from the vacuole in yeast in response to hypertonic shock [16]. It has been documented that CA has great potential to activate Yvc1 in mammalian cells [17]. We previously found that CA was able to disturb $Ca^{2+}$ homeostasis in *P. capsici*. This drove us to think about the possibility for the regulation of *P. capsici* $Ca^{2+}$ channels by CA. On the other hand, it is well known that the RNA-seq has been used a routine method for studying the fungal growth, pathogenesis, and the genes involved in resistance [18–20].

In this work, we first identified the antifungal effect of CA against *P. capsici* in vivo and in vitro. Then, we studied the role of PcNCX1 (a sodium/calcium exchanger) in CA-disturbed $Ca^{2+}$ homeostasis in *P. capsici* based on yeast complementation experiments. In addition, we analyzed the gene expression profile of CA-treated *P. capsici* using the RNA-seq. The involvement of PcNCX1-$Ca^{2+}$ in the antifungal mechanism for CA against *P. capsici* was discussed as well.

## 2. Materials and Methods

### 2.1. Strains and Growth Conditions

*P. capsici* strain (Institution of plant protection, Nanjing Agricultural University) was cultured on PDA (potato dextrose agar) medium at 25 °C, For liquid cultures, *P. capsici* strains were grown in a V8 medium at 25 °C at 100 rpm.

### 2.2. Complementation of the S. cerevisiae ΔYvc1

The CDS fragment of *PcNCX1* (BT032616.1, Table S1) was amplified from *P. capsici* RNA by PCR using primer pairs Ncx1-F (Hind III): aagcttATGGCTCAGCGCCGACGC and Ncx1-R (Xho I): ctcgagTTAAATGATGTCCAAGCCGC. The fragment was purified and cloned into a pMD19-T vector (Takara, Co., Dalian, China) to generate plasmid pMD-*PcNCX1*. The *PcNCX1* fragment was digested with HindIII-XhoI from pMD-*PcNCX1* and subcloned into the pYES2 yeast expression vector digested with HindIII-XhoI to generate pYES2-*PcNCX1* expressing *PcNCX1* under the Gal1 promoter. After verification by sequencing, the pYES2-*PcNCX1* plasmid was introduced into the *S. cerevisiae* ΔYvc1 mutant strain YOR087W (BY4741; MATa; his3Δ1; leu2Δ0; met15Δ0; ura3Δ0; YOR087w::kanMX4), and

it was purchased from Euroscarf (accessed on 15 May 2022, http://www.euroscarf.de/search.php?search=YOR087W, Germany). Yeast cells were incubated on a liquid YPD (yeast extract peptone dextrose) medium containing CA at various concentration.

### 2.3. Inoculation of P. capsici on Pepper Seedlings and Plants

Zoospores of *P. capsici* were induced from 5-day-old sporangia by washing with sterile distilled water for 24 h at 25 °C and were adjusted to a concentration of $1 \times 10^5$ spores/mL for inoculations. Pepper (*Capsicum annuum*) fruits (Sujiao 5) were surface-sterilized with 1% NaClO for 10 min followed by washing with double-distilled water (ddH$_2$O) for three times. A total of 50 μL of zoospores suspension ($1 \times 10^5$ spores/mL) with 1 mM CA [21] were inoculated on the surface of each pepper fruit at three different sites. The inoculated pepper fruits were placed in a plastic box with a water-saturated filter paper and were incubated at 25 °C. The disease symptoms were observed at 4 days post inoculation (dpi). For pepper plants inoculation, a total of 50 μL of zoospores suspension ($1 \times 10^5$ spores/mL) with 1 mM CA was inoculated at the stem base of 35-day-old pepper plants (Sujiao 5). The inoculated plants were kept in a chamber with photosynthetic active radiation of 200 μmol/m$^2$/s, a photoperiod of 12 h and a temperature at 25 °C for 4 days followed by the observation of disease symptoms.

### 2.4. Histochemical Staining

Trypan blue staining was used to evaluated cell death in the root pepper plant. The pepper roots at 4 dpi were carefully washed with ddH$_2$O. Then, the roots were stained with Trypan blue (10 mg/L) for 1 h and detained in lactophenol for 1 h followed by photographing with a digital camera [22,23].

The intracellular Ca$^{2+}$ in yeast cells were labeled in situ with specific fluorescent probe Fluo-3-AM (Beyotime Biotechnology Institute, Haimen, China). Fluo-3-AM was added to yeast suspension at a final concentration of 150 μM. Then, the suspension was incubated at 37 °C for 1 h. Confocal microscopy was performed using a Zeiss Axiovert 200 M microscope equipped with a Zeiss LSM 710 META system using 940/1.2 NA and 963/1.2 NA C-Apochromat water immersion objectives. Images were acquired and processed using LSM 710 AIM version 4.2 SP1 software (Zeiss, Oberkochen, Germany).

### 2.5. Determination of Glycerol Content

The *P. capsici* strain grown in liquid V8 media were treated with 0, 0.8, 1.6, and 2 mM of CA [21] for 24 h. Then, the mycelia were harvested for the determination of glycerol content. Glycerol content was determined using cupric glycerinate colorimetric assay under the instructions of a glycerol detection kit (E1012; Beijing Applygen Technologies Inc., Beijing, China).

### 2.6. Determination of Malondialdehyde (MDA) Content

The content of MDA was determined as an indicator of the lipid peroxidation in CA-treated mycelia of *P. capsici*. An MDA detection kit (A003; Nanjing Jiancheng Bioengineering Institute, Nanjing, China) was selected to determine the MDA's level based on the spectrophotometric determination of the reaction between MDA and 1,3-diethyl-2-thiobarbituric acid (TBA) assisted by trichloroacetic acid (TCA).

### 2.7. Scanning Electron Microscopy (SEM)

The *P. capsici* mycelia was cultivated with liquid V8 with or without CA (1 mM). The samples were fixed in 2.5% glutaraldehyde for 2 h. Then, the samples were transferred to alcohol with increasing concentrations (30%, 50%, and 70%) for 15 min followed by 20 min in 80%, 90%, 95%, and 100% of alcohol, consecutively. The graded aqueous series of acetone (25%, 50%, 75%, and 100%) was critical point dried with CO$_2$ using acetone as an intermediate fluid. After that, the fungal samples were visualized and photographed by SEM (EVO-LS10, ZEISS, Jena, Germany).

*2.8. qRT-PCR Analysis*

Quantitative RT-PCR was performed using the ABI 7500 Fast Real-Time PCR System (ABI Co., Carlsbad, CA, USA) and transcripts were analyzed using the 7500 System SDS software (ABI). To compare the relative abundances of target gene transcripts, the average threshold cycle (Ct) was normalized to that of *Actin* (Actin-F: ACTGCACGTTCCA-GACGATC; Actin-R: CCACCACCTTGATCTTCATG) for each of the treated samples as $2^{-\Delta Ct}$, where $-\Delta Ct = (C_{t,\ target\ gene} - C_{t,\ actin})$. Fold change was calculated as $2^{-\Delta\Delta Ct}$, where $-\Delta\Delta Ct = (C_{t,\ target\ gene} - C_{t,\ actin})_{treated\ with\ CA} - (C_{t,\ control} - C_{t,\ actin})_{control}$. The primers were 131120-RT-F: ACTCGCTCTGGTATACGTGG and 131120-RT-R: AACAACAGTTTAC-CACCCGC.

*2.9. Total RNA Isolation and Library Preparation for Transcriptome Sequencing*

Total RNA was extracted from *P. capsici* mycelia treated with 1 mM CA for 24 h using Trizol reagent (Invitrogen, Carlsbad, CA, USA) following the manufacture's protocol. *P. capsici* mycelia without CA were taken as control. The quality and integrity of RNA was evaluated using Bioanalyzer 2100 (Agilent Technologies, CA, USA) and agarose gel electrophoresis. A total amount of 3 μg RNA per sample was used as input material for the RNA sample preparation. Sequencing libraries were generated using NEBNext Ultra$^{TM}$ RNA Library Prep Kit for Illumina (NEB, Ipswich, MA, USA) following the manufacturer's recommendations. The mRNA was purified from total RNA using oligo (dT)-attached magnetic beads. The first strand cDNA was synthesized using random hexamer primer and M-MuLV reverse transcriptase. Second stand cDNA syntheses were subsequently performed using DNA polymerase I and RNase H. After second-strand cDNA synthesis and dual-index adaptor ligation, the paired-end library fragments with 150–200 bp in length were purified with AMPure XP system (Beckman Coulter, Beverly, CA, USA). The library quality was assessed using Agilent Bioanalyzer 2100 system. Then, four libraries (Control-1, -2 and CA-treatment-1, -2) preparations were sequenced on an Illumina Hiseq 2000 platform to generate 100 bp paired-end reads (BGI, Shenzhen, Guangzhou, China).

*2.10. Quality Control, Transcriptome Assembly and Differential Expression Analysis of Unigenes*

For quality control, raw sequencing data were filtered by Fast QC software, and low-quality reads were filtered out. Both clean reads were de novo assembled with Trinity with the default settings, with the exception that min_kmer_cov was set to 2 by default.

The analysis of DEGs (differential expression genes) between two samples were performed with the DEGseq R package. The *p*-value was adjusted using the q value. The q value < 0.005 and the absolute value of $\log_2 > 1$ were set as the threshold for significant difference in gene expression. Gene ontology (GO) analyses of DEGs were implemented by BGI WEGO (Web Gene Ontology Annotation Plotting, accessed on 15 August 2015, http://wego.genomics.org.cn/cgi-bin/wego/index.pl) and agriGO (GO Analysis Toolkit and Database for Agricultural Community, accessed on 15 August 2015, http://bioinfo.cau.edu.cn/agriGO/index.php). Pathways were significantly enriched with KEGG (Q-value < 0.05).

*2.11. Statistical Analysis*

For the data of the measurement of physiological parameters, each result was presented as the mean and standard deviation (SD) of three replicates. The least significant difference (LSD) test was performed on all data following a one-way analysis of variance (ANOVA) to test for significant differences ($p < 0.05$) among different treatments.

**3. Results**

*3.1. CA Controlled the Pepper Phytophthora Blight*

In our previous study, CA exerted efficient inhibitory effects on both mycelial growth ($EC_{50} = 0.75$ mM) and zoospore germination (MIC = 0.4 mM) of *P. capsici* [6]. To investigate the role of CA in the pathogenicity of *P. capsici*, CA (1 mM) was applied on pepper fruit

surface inoculated with *P. capsici* zoospore suspensions. Pepper fruit inoculated with *P. capsicia* showed severe disease symptoms while treatment with CA resulted in much fewer fruit rots at 4 dpi (Figure 1A). Similar results were obtained with pepper plants (Figure 1B). Inoculation with *P. capsici* at the base of stem resulted in the infection of stem and roots, further leading to seedling lodging and leaf withering. Treatment with CA was able to rescue seedling growth from the infection of *P. capsici* (Figure 1B). As a kind of soil borne disease fungus, *P. capsici* can infect plant roots to cause root cell death. We analyzed root cell death using trypan blue staining. CA remarkably attenuated root cell death of seedlings inoculated with *P. capsici* (Figure 1C). These results indicated that CA effectively controlled the infection of *P. capsici* on pepper.

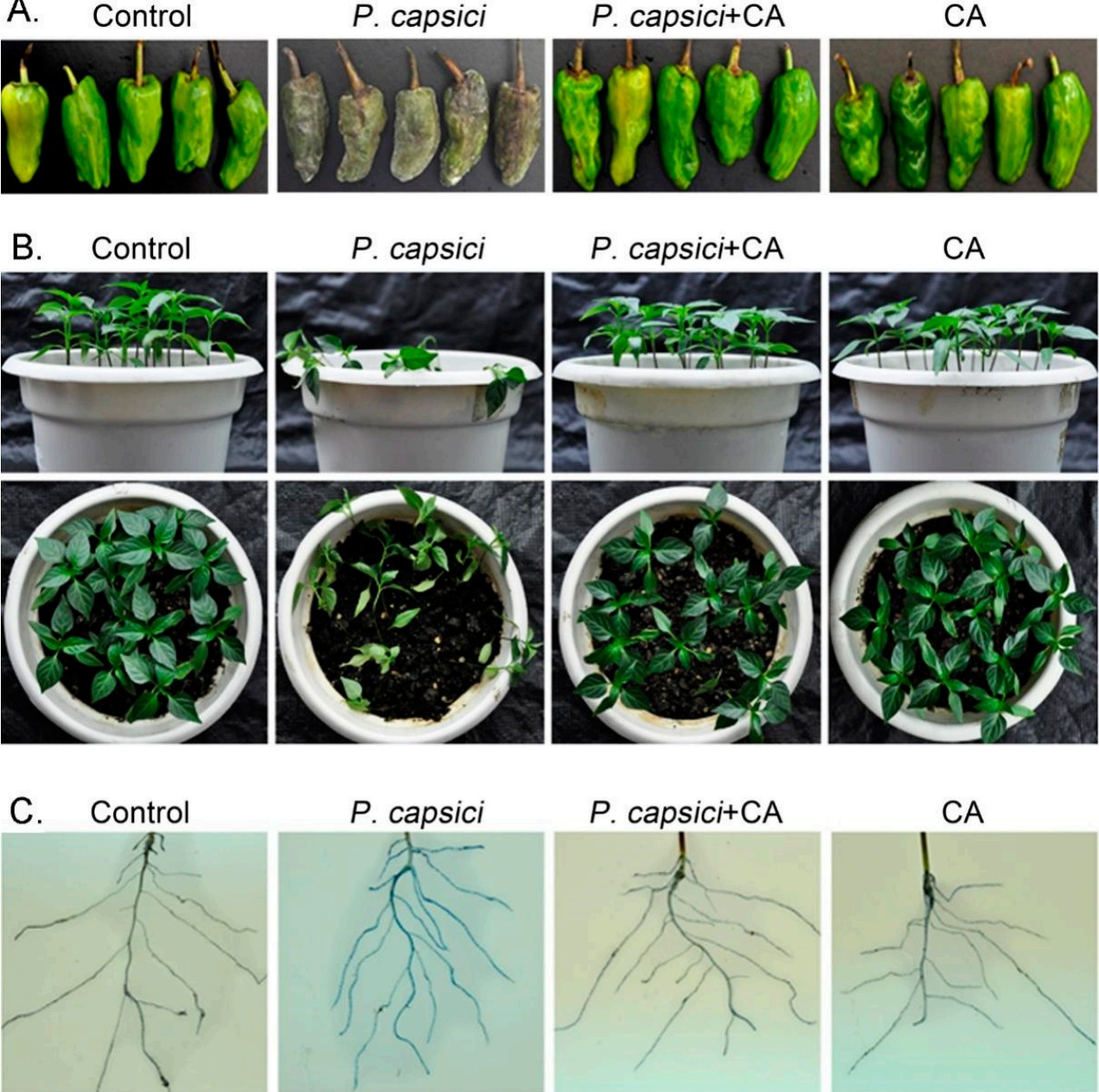

**Figure 1.** Inhibition of Phytophthora blight by CA on pepper seedling and fruit/ (**A**) Inhibition effect of Phytophthora blight by CA (1 mM) on pepper fruit and pepper seedling after 3 h inoculation with zoospores suspension ($1 \times 10^5$ spores/mL). (**B**) Pepper seedlings were treated with zoospore suspensions supplemented with or without 1 mM CA. (**C**) Trypan blue staining with pepper roots treated with zoospore suspensions supplemented with or without 1 mM CA.

### 3.2. CA Impacted Cell Integrity of P. capsici

Results from the SEM revealed clear morphological alterations in the hyphae of *P. capsici* under CA treatments. Compared to the control group, collapsed and fragmented hyphae were observed in mycelia treated with CA (Figure 2A). The accumulation of MDA is a typical indicator of oxidative injury and plasma membrane damage [24]. Treatment with CA led to significant increase in MDA content in the mycelia of *P. capsici* in a dose-dependent manner (Figure 2B). The intercellular glycerol plays a critical role in the protection of fungal cells from osmotic stress [25]. We found that treatment with CA resulted in a significant decrease in intercellular glycerol content in the mycelia of *P. capsici.* The changes of MDA and glycerol content also confirmed the loss of cell integrity in the hyphae of *P. capsici* under CA treatment.

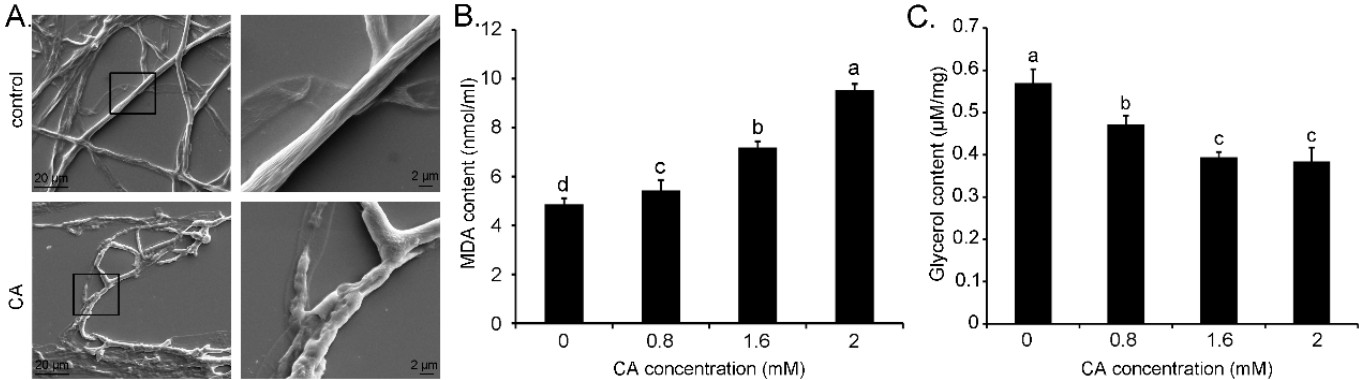

**Figure 2.** The defect in cell wall integrity of mycelia treated with CA/ (**A**) SEM observation of the hyphal morphology of *P. capsici* treated with 1 mM CA; (**B,C**) MDA content and glycerol content of the mycelia of *P. capsici*. Mycelia treated with CA at 0, 0.8, 1.6, and 2 mM for 24 h were harvested for the measurements. Each value is presented as the average of three replicates. Letters indicates that mean values are significantly different ($p < 0.05$) between the treatment and the control.

### 3.3. Identification of CA-Regulated Ca$^{2+}$ Channels in P. capsici

We previously found that CA induced immediate Ca$^{2+}$ efflux from zoospores of *P. capsici* [9]. In this study, the yeast mutant Δ*Yvc1* was used to investigate CA-regulated Ca$^{2+}$ channel in *P. capsici.* Δ*Yvc1* was more tolerant to CA compared to the wild type (WT) yeast, indicating that Yvc1 played a role in response to CA (Figure 3A). Then, we complemented the yeast mutant Δ*Yvc1* with *PcNCX1* (the *P. capsici* sodium/calcium exchanger). The transformants (Δ*Yvc1/NCX1* #1 and Δ*Yvc1/NCX1* #2) expressing *PcNCX1* showed recovered sensitivity to different concentration of CA (Figure 3A), indicating that *PcNCX1* maybe the potential target of CA. Then we tested the intracellular Ca$^{2+}$ level using specific fluorescent probe Fluo-3 AM. In CA-free medium, the intracellular Ca$^{2+}$ was successfully labeled with green fluorescence in WT yeast, the deletion mutant Δ*Yvc1*, and the complemented strain Δ*Yvc1/PcNCX1* (Figure 3B). Compared to WT and the Δ*Yvc1/PcNCX1* strains, the deletion mutant Δ*Yvc1* stilled showed strong fluorescent signal upon CA treatment (Figure 3C). These results showed that the PcNCX1 can partly restore CA-induced Ca$^{2+}$ efflux in the yeast deletion mutant Δ*Yvc1*.

### 3.4. Transcriptome Analysis of P. capsici Treated with CA

To explore the mechanisms underlying the inhibition of *P. capsici* by CA, we performed RNA-sequencing of *P. capsici* hyphae under CA treatments and found that the hyphae are thin compared to control (Figure 4A). Approximately 46 million (96.81%) and 44 million (98.33%) clean reads were obtained from CA-treatment and control, respectively (Table 1). Among all the clean reads, approximately 91.46% and 91.32% of the CA-treatment and control were mapped to the reference genome, respectively (Table 1). We identified 956 up-regulated unigenes and 2575 down-regulated unigenes upon CA treatment (Figure 4B).

Enrichment analysis was performed to illustrate the biological functions of the identified DEGs. In total, 607, 1184, and 1198 DGEs of cellular components, molecular function, and biological process were enriched in 37 (gene ontology) GO terms. Five GO terms including "metabolic process" (805), "Cellular process" (578), and "Single-organism process" (457) in the category of biological process; "Catalytic activity" (1229) and "Transporter activity" (215) in the category of molecular function were significantly enriched in the CA-treatment sample (Figure 5A). In addition, 1591 DEGs were enriched in KEGG pathways. The most enriched pathways were "Metabolic pathways" (388) and "Biosynthesis of secondary metabolites" (169) (Figure 5B).

$Ca^{2+}$ acts as a second messenger to modulate numerous intrinsic metabolic processes by regulating a set of $Ca^{2+}$-binding proteins [26]. CA affected $Ca^{2+}$ homeostasis by regulating PcNCX1. We screened the expression level of genes coding for $Ca^{2+}$-regulated proteins from RNA-sequencing data. As expected, in the CA-treatment samples, 25 and 5 calcium-related proteins were down- and up-regulated, respectively (Table 2). This may due to decreased intracellular $Ca^{2+}$ levels in CA-treated *P. capsici*. In addition, we found that the expressions of four $Ca^{2+}$ pump genes (111,992, 109,877, 126,140, and 549,419) were also down-regulated by CA, indicating that the down-regulation of these genes may also contributed CA-suppressed intracellular $Ca^{2+}$ levels in *P. capsici*.

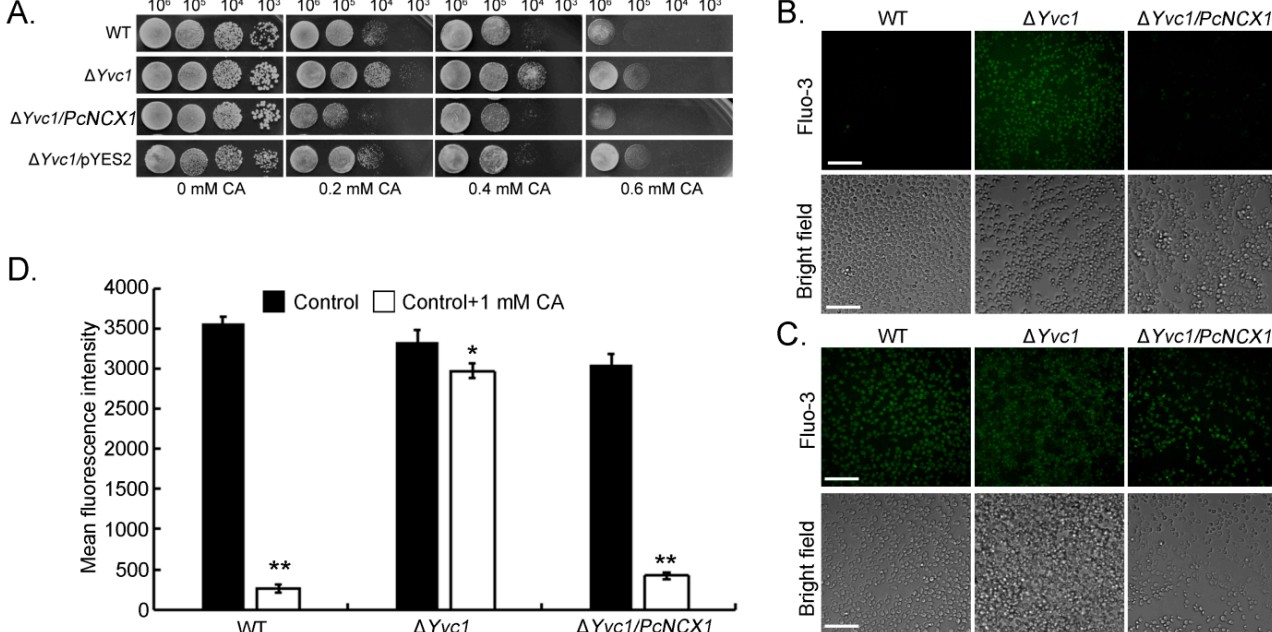

**Figure 3.** The *PcNCX1* gene rescued the CA sensitivity of *yeast ΔYvc1* mutant. (**A**) The yeast *ΔYvc1* mutant was transformed with the pYES2-*PcNCX1* construct expressing PcNCX1. The wild type strain was also transformed with the empty pYES2 vector as a control. Serial dilutions of cultures of all strains were grown overnight on SD-Met-Leu-His (Synthetic dropout medium lacking of methionine, leucine, and histidine) (glucose) or SD-Met-Leu-His (glucose + 0.2/0.4/0.6 mM CA) plates, and grown at 30 °C for 4 days and photographed. The complementary transformant restores the immediate $Ca^{2+}$ efflux of the yeast *ΔYvc1* mutant treated with (**B**) and without (**C**) CA. (**D**) The mean fluorescence intensity of the wild type, the *ΔYvc1* mutant and the complementary transformant *ΔYvc1/NCX1* treated with 1 mM CA. The experiment was repeated at least three times, and representative results were photographed. Asterisk indicates statistically significant differences (*, $p < 0.05$; **, $p < 0.01$), Bar = 30 μm.

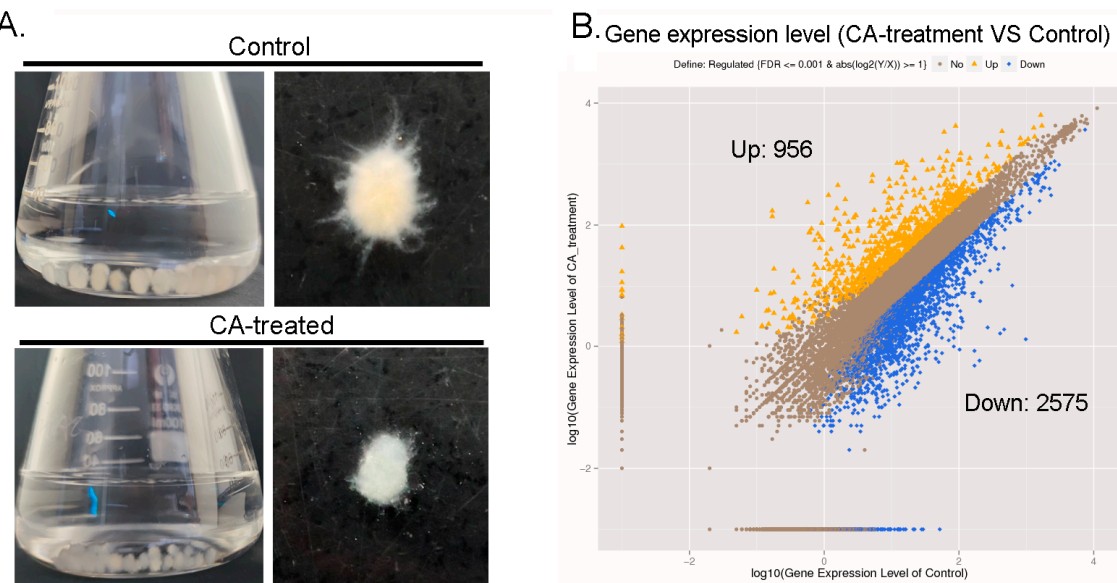

**Figure 4.** The summary of RNA-Seq of the sample treated with CA. (**A**) The mycelia plug of the *P. capsici* treated with 1 mM CA, and the mycelia plugs were cultured with V8 medium and transferred to ddH$_2$O during photography. (**B**) The distribution of DEGs. Yellow spots represent up-regulated DEGs and blue spots indicate down-regulated DEGs. Those shown in brown are unigenes that did not show obvious changes.

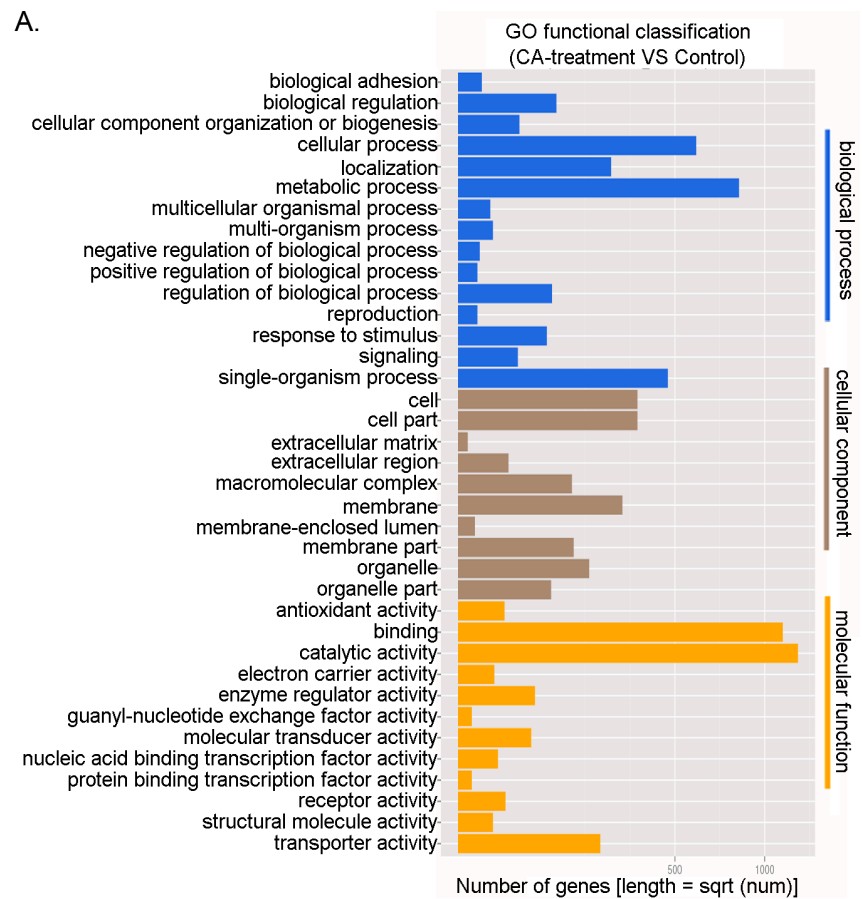

**Figure 5.** *Cont.*

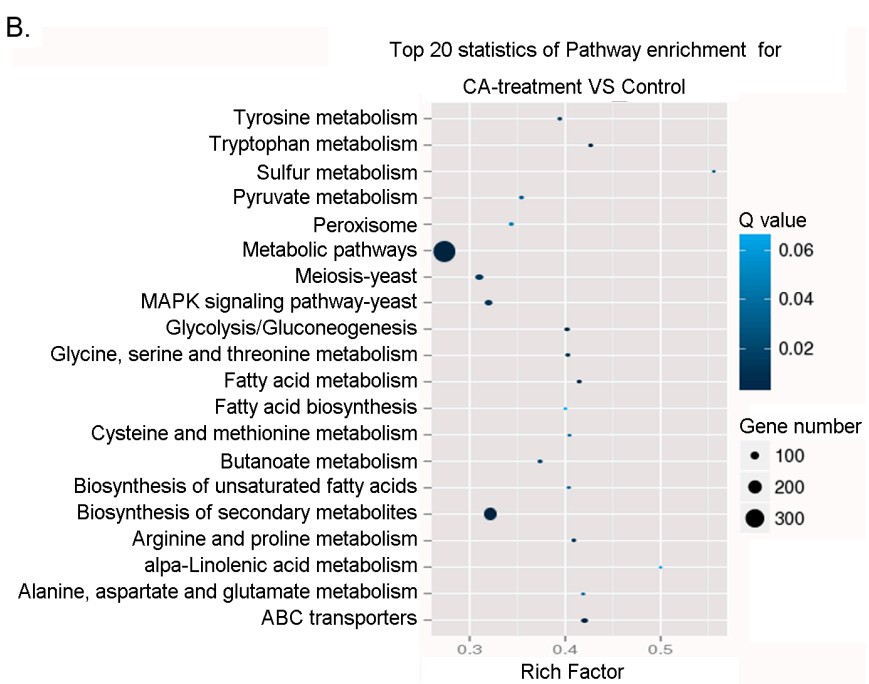

**Figure 5.** Histogram of gene ontology (GO) classification (**A**) and Scatter diagram of KEGG pathway enrichment (**B**).

**Table 1.** Summary of sequences analysis of sample of treated with CA and control.

| Sample Name | Raw Reads | Clean Reads | Genome Map Rate | Gene Map Rate | Expressed Transcripts |
|---|---|---|---|---|---|
| CA-treatment | 47,695,094 | 46,172,436 | 91.46% | 71.95% | 13,230 |
| control | 45,423,808 | 44,664,188 | 91.32% | 68.05% | 13,667 |

**Table 2.** Summary of DEGs encoding calcium-related genes.

| Gene ID | Log2 Ratio (CA_Treatment/Control) | Up/Down-Regulation (CA_Treatment/Control) | $p$-Value | Gene Annotation |
|---|---|---|---|---|
| 534,503 | 2.100603941 | Up | $1.48 \times 10^{-13}$ | Probable calcium-binding protein |
| 534,501 | 1.757530286 | Up | $1.04 \times 10^{-15}$ | Putative calmodulin-3 |
| 507,044 | 1.698515183 | Up | 0 | Calmodulin |
| 548,475 | 1.629408523 | Up | $4.67 \times 10^{-172}$ | Calnexin homolog |
| 535,137 | 1.181965909 | Up | $1.34 \times 10^{-27}$ | Mitochondrial proton/calcium exchanger protein |
| 68,193 | −11.45121111 | Down | $1.46 \times 10^{-6}$ | Calcium-dependent protein kinase 9 |
| 111,992 * | −6.434628228 | Down | $6.64 \times 10^{-26}$ | Plasma membrane calcium-transporting ATPase 4 |
| 130,774 | −5.933690655 | Down | $4.95 \times 10^{-17}$ | Caltractin |
| 507,496 | −5.36862133 | Down | $2.23 \times 10^{-11}$ | Probable calcium-binding protein CML13 |
| 6853 | −4.37625161 | Down | $3.55 \times 10^{-20}$ | EF-hand calcium-binding domain containing protein 10 |
| 123,091 | −2.64385619 | Down | $3.75 \times 10^{-12}$ | Calcium-dependent protein kinase 7 |
| 109,877 * | −2.485426827 | Down | $6.44 \times 10^{-5}$ | Calcium-transporting ATPase 10, plasma membrane-type |
| 117,823 | −2.37469857 | Down | $2.46 \times 10^{-13}$ | Calcineurin subunit B |
| 6396 | −2.348262243 | Down | $7.47 \times 10^{-13}$ | Probable calcium-binding protein CML45 |
| 109,832 | −2.270282489 | Down | $9.25 \times 10^{-27}$ | Uncharacterized calcium-binding protein C50C3.5 |
| 98,673 | −2.159568209 | Down | $9.27 \times 10^{-35}$ | Calretinin |
| 121,662 | −2.137074852 | Down | $2.85 \times 10^{-9}$ | Calcineurin B-like protein 2 |
| 531,031 | −1.941557727 | Down | $1.38 \times 10^{-63}$ | Probable calcium-binding protein CML18 |
| 119,329 | −1.910916456 | Down | $2.60 \times 10^{-8}$ | Calmodulin-like protein 12 |
| 540283 | −1.893563064 | Down | $1.44 \times 10^{-14}$ | Calcineurin B-like protein 10 |
| 41,424 | −1.816842742 | Down | $2.15 \times 10^{-10}$ | Probable calcium-binding protein CML18 |
| 574,424 | −1.802060622 | Down | $6.45 \times 10^{-22}$ | Voltage-dependent L-type calcium channel subunit alpha-1C |
| 126,140 * | −1.729910837 | Down | $1.77 \times 10^{-7}$ | Plasma membrane calcium-transporting ATPase 4 |
| 525,961 | −1.601016081 | Down | $2.76 \times 10^{-19}$ | Cobalt/magnesium transport protein CorA |
| 41,359 | −1.34356494 | Down | $1.75 \times 10^{-34}$ | Caltractin |
| 547,508 | −1.296891544 | Down | $2.53 \times 10^{-17}$ | Calcium-dependent protein kinase 34 |
| 504,561 | −1.295161185 | Down | $1.48 \times 10^{-9}$ | Calcineurin subunit B |
| 549,419 * | −1.184730681 | Down | $2.02 \times 10^{-12}$ | Plasma membrane calcium-transporting ATPase 4 |
| 542,784 | −1.145992514 | Down | $5.49 \times 10^{-56}$ | Calcium-dependent protein kinase 22 |
| 554,361 | −1.073761931 | Down | $3.95 \times 10^{-18}$ | Caltractin ICL1e |

Note: Asterisk indicates four $Ca^{2+}$ pump proteins.

ABC (ATP-binding cassette) transporters mainly use the hydrolysis of ATP to fuel the transmembrane's transport and participates in the physiological process of drug resistance by facilitating drug efflux [27]. In this study, we found that 45 DEGs (including 34 down-regulated and 11 up-regulated) encoded for ABC transporters and drug-resistance proteins in the CA-treated samples (Table S2). These results suggested that ABC transporters in *P. capsici* may be differentially regulated by CA.

## 4. Discussion

*P. capsici* is a highly destructive plant pathogen worldwide. CA is a widely used flavoring agent that has been used to improve food quality. In this study, we showed that CA was able to destroy the cell integrity of *P. capsici* hyphae and to control pepper phytophthora effectively. Yvc1 in *S. cerevisiae* is a vacuolar membrane localized calcium channel. Yvc1 mediates $Ca^{2+}$ release from the vacuole in response to hypertonic shock [28–30]. We found that PcNCX1 can partly restore the immediate $Ca^{2+}$ efflux of the yeast deletion mutant $\Delta Yvc1$ under CA treatment. Thus, CA could target PcNCX1 to cause cell defects of *P. capsici*.

CA could be a safe antibacterial and antifungal alternative due to its food additive properties. Several studies have shown that CA can inhibit the growth, morphology, and mycelial formation of *C. albicans*. Our previous study indicated that CA can control the pepper blight by inhibiting the zoospore germination and vegetable growth of *P. capsici* [9]. In this study, we found that CA significantly controls the pepper Phytophthora blight by destroying the cell wall integrity of *P. capsici* (Figures 1 and 2). Cell-wall integrity plays a vital role in fungal growth and development as well as the ability to survive under stress conditions [31]. We found collapsed and fragmented hyphae in *P. capsici* mycelia treated with CA (Figure 2A). In addition, MDA concentration and intercellular glycerol measured tests showed that the hyphae of *P. capsici* treated with CA was more sensitive to oxidative stress and osmotic stress (Figure 2B). The pathogenicity of *M. oryzae* is dependent on several key genes (e.g., *MoSWI6*, *MoRGS1*, and *MoPDEH*) modulating cell wall integrity [32–34]. It is reported that CA can affect the synthesis of 1,3-β-D-glucans of *Aspergillus fumigatus* and can inhibit chitin synthase 1 in *Saccharomyces cerevisiae* [35]. RNA-seq analysis showed that the expressions of twelve glucan 1,3-β-glucosidase of *P. capsici* treated with CA were significantly different from control (Table S3), which suggested that CA affected cell-wall integrity by regulating the expression of glucan 1,3-β-glucosidase.

$Ca^{2+}$ is a second messenger involved in the response to different stress signals (e.g., hypotonic, hypertonic shock, and nutrients) in yeast [36]. $Ca^{2+}$-binding proteins interact with specific transcription factors to modulate the cell-wall's integrity and pathogenicity of fungi [37,38]. $Ca^{2+}$ concentration is regulated precisely by the endoplasmic reticulum (ER), Golgi, and vacuole [39]. Yvc1 is involved in the regulation of cytoplasmic $Ca^{2+}$ homeostasis [40]. In this study, we transformed *PcNCX1* (a sodium/calcium exchanger) to the yeast mutant $\Delta Yvc1$. The transformants expressing *PcNCX1* showed restored sensitivity to CA (Figure 3A), coinciding with the decrease in intracellular $Ca^{2+}$ level in complemented strain $\Delta Yvc1/PcNCX1$ upon CA treatment (Figure 3B,C). In *Candida albicans*, CA introduced apoptosis through accumulating the intracellular reactive oxygen species and calcium in the cytoplasm [41]. In *Brassica rapa*, CA can be able to prime plant defense by regulating endogenous $Ca^{2+}$ in order to facilitate cadmium tolerance [42]. In our pervious study, we have found that CA inhibited the growth of *P. capsici* by stimulating $Ca^{2+}$ efflux [9]. Combining with the data in this study, it can be suggested that CA may regulate *PcNCX1* to induce the decrease in intracellular $Ca^{2+}$ levels, further leading to the growth inhibition of *P. capsici*.

The Yeast complementation experiment suggested that PcNCX1 is a potential target of CA. Thus, we suspect that CA may directly interact with PcNCX1 to modulate its activity. CA has been identified as an agonist of hTRPA1 (human transient receptor potential ankyrin 1). CA directly binds to a set of cysteine residues in the N-terminus of hTRPA1, leading to covalent protein modification. These modifications may activate the channel's activity by altering subunit–subunit interactions or by modulating the association of hTRPA1 with other cellular proteins [43]. Whether CA can modulate the channel activity of PcNCX1

through similar mechanisms needs further studies. In addition, CA can modulate mammalian TRPs at a transcriptional level [44]. In this study, we found that CA down-regulated the expression of two $Ca^{2+}$ pump genes in *P. capsici* as well. Therefore, CA may modulate $Ca^{2+}$ flux by regulating $Ca^{2+}$ channels at transcriptional level.

The RNA-seq of *P. capsici* treated with CA was analyzed, and we found that 956 up-regulated and 2575 down-regulated genes in the CA-treated sample and the functions of these DEGs varied (Figure 4B). However, we could not detect *PcNCX1* in the RNA-seq data. Moreover, we investigated the expression of *PcNCX1* in the sample treated with CA with qRT-PCR and found that the expression level was significantly down-regulated (Figure S1). In addition, most of the calcium-related genes (25/30) were down-regulated in CA-treated sample (Table 2), suggesting that CA could introduced decreased intracellular $Ca^{2+}$ level in *P. capsici*. The chemical insensitivity may include non-specific efflux pumps and detoxification, which can be performed by proteins such as the ABC transporter and the cytochrome P450 protein, respectively [45]. CA treatments resulted in the downregulation of most ABC transporter genes (34/45) in *P. capsici*, suggesting that CA shows strong antifungal activities by possibly suppressing the detoxification system of *P. capsici*. In our next research, we will study the function of these $Ca^{2+}$ transporter associated genes using the CRISPER-CAS9 method, which further enrich the action mechanism of CA to *P. capsici*.

## 5. Conclusions

CA effectively control the pepper Phytophthora blight caused by *P. capsici*. The loss of cell-wall integrity is closely related to the antifungal effect of CA against *P. capsici*. Yeast complementation experiments showed that the sodium/calcium exchanger PcNCX1 restored the immediate $Ca^{2+}$ efflux of $\Delta Yvc1$. Our study demonstrates that PcNCX1-mediated $Ca^{2+}$ efflux may be one of the important reasons to explain the antifungal effect of CA against *P. capsici*, which provides crucial insights into the antimicrobial action of CA.

**Supplementary Materials:** The following supporting information can be downloaded at: https://www.mdpi.com/article/10.3390/agronomy12081763/s1, Figure S1: The expression of PcNCX1 in the sample treated with CA by qRT-PCR. Error bars represent standard deviation; Table S1: The CDS sequence of PcNCX1; Table S2: Summary of DEGs encoding ABC transporter genes; Table S3: Summary of DEGs encoding glucan 1,3-beta-glucosidase genes.

**Author Contributions:** Writing—first draft preparation, Z.Q., J.C., M.L. and H.L.; Writing—review and editing, Z.Q., J.C., M.L., Y.W., L.Z., Z.S. and C.X.; funding acquisition, J.C.; figures and tables, Z.Q. and L.L. All authors have read and agreed to the published version of the manuscript.

**Funding:** This study was funded by Jiangsu Agricultural Science and Technology Innovation fund (CX(20)1011).

**Institutional Review Board Statement:** Not applicable.

**Informed Consent Statement:** Not applicable.

**Data Availability Statement:** Data sharing not applicable.

**Conflicts of Interest:** The authors declare no conflict of interest.

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
