# Peer review of "The Sodium/Calcium Exchanger PcNCX1-Mediated Ca2+ Efflux Is Involved in Cinnamaldehyde-Induced Cell-Wall Defects of Phytophthora capsici"

_agronomy, doi:10.3390/agronomy12081763_

Round 1

Reviewer 1 Report

The current study investigated the role of a sodium/calcium exchanger, PcNcx1 in disintegration of Phytophthora capsici cell wall induced by cinnamaldehyde (CA) through several methods including RNASeq.

Mode of action determination is crucial for development of any type of fungicides. This study provided proves on induction of cell wall disintegrity upon CA application, however, the major drawback is the lack of evidence for involvement of PcNcx1 in Phytophthora capsici upon CA treatment. The RNASeq didn't report any change, and the mutation study has not been performed in P. capsici. Therefore, the manuscript is recommended for rejection based on the current experimental design/methodologies, presented results and unsupported conclusions. Please find some comments about the paper below.

1) Add information about RNASeq benefits to the fungal studies and why is it being used.

2) Line 54, spell out MIC.

3) Line 55-56, remove this sentence "we still ... in agriculture". Because you have already reported some knowledge about it.

4) Line 70, change reference 15 to something related to fungi or at least plants not the mammals. Also, in the entire study, there are several examples from mammals studies. Please try to replace those literatures to some corresponding to fungal or at least plants literatures.

5) Line 86, this is a critical point of the experimental design that why didn't the author use the P. capsici mutants? Application of the S. cerevisiae for the trial is fine, but for the proving the objectives of the study, the entire processes used in the methods should be applied to P. capsici, not any other fungus.

6) Line 87, either add the sequence of the PcNcx1 to the supplementary files or provide the accession number from NCBI database.

7) Line 147, how old (days-post-treatment with CA) were the mycelia used for RNA extraction? 

8) Line 148, since 1mM concentration of CA was used in the RNASeq experiment, please add the picture of the culture supplemented with 1mM CA to the figure 3A to compare this concentration with others.

9) Line 158, were the adaptor used in the library preparation with single index or dual index? Please add to the text.

10) Lines 165-168 belongs to results and should be transferred to there. Also, it is more useful to be consistent in the numbers, e.g., present the numeric values as the read counts not bases.

11) Line 229 and 327, replace sensibility to sensitivity.

12) Lines 238, enhance the quality of figure 3A,B,C. In addition, in the figure 3 caption, add the days-post-treatment to the legend A. Fix the order of legends B and C.

13) Lines 251-252, remove this text and its figure "The Venn diagram ... and control, respectively (figure 4A)". This comparison doesn't represent a meaningful result. 

14) Line 271, add KEGG pathway to the caption of figure 5.

15) Line 279, use this text "the expression of two Ca2+ ..."

16) Line 279, it is stated that two Ca pump genes were found down regulated; how about the other two with Gene ID 109877 and 126140 from Table 2?

17) Line 290, the entire Table 3 should be used as a supplementary table. The text has already provided sufficient information about it.

18) Line 291, the entire discussion should be elaborating more about their research results and compare them with relevant studies, preferably non-mammalian studies. For example, in second paragraph, there is disassociation between the literature discussed and the results of the study.

19) Line 304, italicize Streptococcus.

20) Lines 319-320, since RNASeq is a comprehensive method encompassing global gene expression profile, inspection of a gene regulation upon CA induction should not be a concern. So, please check the DEGs for the presence of 1,3 B-D-glucans and report in the text.

21) Lines 324-325, remove all "the" from these two lines.

22) Line 334, "CA failed to regulate the expression of PcNCX1 (DVH05_027425) in P. capsici". a) This is a major concern in this study. The entire study is about the PcNcx1 role, but this sentence makes the study objectives meaningless. Please justify the goals of the study based on this text. b) Did the RNASeq also fail to show any change in regulation of PcNcx1? c) Why not searching for other potential isoforms/homologs of calcium transporter in your RNASeq data?

23) Lines 336-343, to elaborate involvement of other pathways in calcium transport upon CA treatment, please add fungal, bacterial or plant related studies.

24) Line 337, replace "a" with "an".

25) Line 349, use "most of the".

26) Line 357, the conclusion is not supported by the data presented. 

Author Response

The current study investigated the role of a sodium/calcium exchanger, PcNcx1 in disintegration of Phytophthora capsici cell wall induced by cinnamaldehyde (CA) through several methods including RNASeq.

Mode of action determination is crucial for development of any type of fungicides. This study provided proves on induction of cell wall disintegrity upon CA application, however, the major drawback is the lack of evidence for involvement of PcNcx1 in Phytophthora capsici upon CA treatment. The RNASeq didn't report any change, and the mutation study has not been performed in P. capsici. Therefore, the manuscript is recommended for rejection based on the current experimental design/methodologies, presented results and unsupported conclusions. Please find some comments about the paper below.

1) Add information about RNASeq benefits to the fungal studies and why is it being used.

Response: We have added the section with “RNA-sequcing has been demonstrated to play critical role in the fungal growth, pathogenesis and the genes involved in resistance [18-20]. Thus, the gene expression profile of CA-treated P. capsici to gain some insight into its mode of action.” (line 81-83)

2) Line 54, spell out MIC.

Response: We have spell out MIC with “Minimum inhibitory concentrations” (line 55).

3) Line 55-56, remove this sentence "we still ... in agriculture". Because you have already reported some knowledge about it.

Response: We have removed the section.

4) Line 70, change reference 15 to something related to fungi or at least plants not the mammals. Also, in the entire study, there are several examples from mammals studies. Please try to replace those literatures to some corresponding to fungal or at least plants literatures.

Response: We have replaced the reference with “Wang, P.; Li, Z.; Wei, J.; Zhao, Z.; Sun, D.; Cui, S. A Na+/Ca2+ exchanger-like protein (AtNCL) involved in salt stress in Arabidopsis. J Biol Chem. 2012, 287, 44062-44070. http://dx.doi.org/10.1074/jbc.M112.351643.” Moreover, we also replaced the reference 28 with “Nguyen, Q.; Kadotani, N.; Kasahara, S.; Tosa, Y.; Nakayashiki, H. Systematic functional analysis of calcium-signalling proteins in the genome of the rice-blast fungus, Magnaporthe oryzae, using a high-throughput RNA-silencing system. Mol Microbiol. 2008, 68, 1348-65. http://dx.doi.org/10.1111/j.1365-2958.2008.06242.x”

5) Line 86, this is a critical point of the experimental design that why didn't the author use the P. capsici mutants? Application of the S. cerevisiae for the trial is fine, but for the proving the objectives of the study, the entire processes used in the methods should be applied to P. capsici, not any other fungus.

Response: We thank the reviewer for the positive comments. At the beginning of the experiment, we tried to establish the gene silencing of P. capsici, but failed. So we tried to do it with yeast mutant complementation. Now we are building a crispr-cas9 knockout system for P. capsici.

6) Line 87, either add the sequence of the PcNcx1 to the supplementary files or provide the accession number from NCBI database.

Response: We have added the sequence of the PcNCX1 in Table S1 and the accession number (BT032616.1) in line 92.

7) Line 147, how old (days-post-treatment with CA) were the mycelia used for RNA extraction? 

Response: we have added the section with “Total RNA was extracted from P. capsici mycelia treated with 1 mM CA for 24 h using Trizol reagent (Invitrogen, Carlsbad, CA, USA) following the manufacture’s protocol.” (line 163-165)

8) Line 148, since 1mM concentration of CA was used in the RNASeq experiment, please add the picture of the culture supplemented with 1mM CA to the figure 3A to compare this concentration with others.

Response: we have cultured the mycelia treated with 1 mM CA and added the picture of the mycelium masses in Figure 4A.

9) Line 158, were the adaptor used in the library preparation with single index or dual index? Please add to the text.

Response: The dual index adaptor was used in the library preparation and we added this section with “the pair-end library fragments with 150-200 bp in length were purified with AMPure XP system (Beckman Coulter, Beverly, USA)” line 174-175.

10) Lines 165-168 belongs to results and should be transferred to there. Also, it is more useful to be consistent in the numbers, e.g., present the numeric values as the read counts not bases.

Response: Thanks for the helpful comments. We have transferred this section in the result and we added raw reads in Table 1, modified the description with “To explore the mechanisms underlying the inhibition of P. capsici by CA, we performed RNA-sequencing of P. capsici hyphae under CA treatment and found that the hyphae are thin compared to control (Figure 4A). Approximately 46 million (96.81%) and 44 million (98.33%) clean reads were obtained from CA-treatment and control, respectively (Table 1)” (line 263-267)

11) Line 229 and 327, replace sensibility to sensitivity.

Response: We have made corrections.

12) Lines 238, enhance the quality of figure 3A,B,C. In addition, in the figure 3 caption, add the days-post-treatment to the legend A. Fix the order of legends B and C.

Response: We have reedited the pictures of Figure 3A, B, C, fixed the order of legends B and C, and added the description of legend A with “The yeast ΔYvc1 mutant was transformed with the pYES2-PcNCX1 construct expressing PcNcx1. The wild type strain was also transformed with the empty pYES2 vector as a control. Serial dilutions of cultures of all strains were grown overnight on SD-Met-Leu-His (Synthetic dropout medium lacking of methionine, leucine and histidine) (glucose) or SD-Met-Leu-His (glucose + 0.2/0.4/0.6 mM CA) plates, and grown at 30°C for 4 days and photographed.” (line 252-257)

13) Lines 251-252, remove this text and its figure "The Venn diagram ... and control, respectively (figure 4A)". This comparison doesn't represent a meaningful result. 

Response: We have removed.

14) Line 271, add KEGG pathway to the caption of figure 5.

Response: We have added.

15) Line 279, use this text "the expression of two Ca2+ ..."

Response: We have modified.

16) Line 279, it is stated that two Ca pump genes were found down regulated; how about the other two with Gene ID 109877 and 126140 from Table 2?

Response: Thanks for the helpful comments. We have added the description of these genes 109877 and 126140 in line 296.

17) Line 290, the entire Table 3 should be used as a supplementary table. The text has already provided sufficient information about it.

Response: We have modified.

18) Line 291, the entire discussion should be elaborating more about their research results and compare them with relevant studies, preferably non-mammalian studies. For example, in second paragraph, there is disassociation between the literature discussed and the results of the study.

Response: Thanks for the helpful comments. We have deleted the section “CA was reported to be effective against biofilms formed by Pseudomonas aeruginosa and Staphylococcus aureus. In addition, CA exhibits antimicrobial activity against Streptococcus mutants biofilm formation by modulating its hydrophobicity, aggregation, acid production, acid tolerance and virulence gene expression [27]. Moreover, CA alone or in combination with antibiotics possess potential efficacy against pathogenic Enterobacteriaceae [28].”. And we have made some modification in the discussion.

19) Line 304, italicize Streptococcus.

Response: We have modified.

20) Lines 319-320, since RNASeq is a comprehensive method encompassing global gene expression profile, inspection of a gene regulation upon CA induction should not be a concern. So, please check the DEGs for the presence of 1,3 B-D-glucans and report in the text.

Response: We agreed the reviewer for the positive comments. We have analyzed the expression of glucan 1,3-β-glucosidase of P. capsici treated with CA (Table S3), and found that twelve genes were significantly different from control. The description were shown in line 330-333.

21) Lines 324-325, remove all "the" from these two lines.

Response: We have removed.

22) Line 334, "CA failed to regulate the expression of PcNCX1 (DVH05_027425) in P. capsici". a) This is a major concern in this study. The entire study is about the PcNcx1 role, but this sentence makes the study objectives meaningless. Please justify the goals of the study based on this text. b) Did the RNASeq also fail to show any change in regulation of PcNcx1? c) Why not searching for other potential isoforms/homologs of calcium transporter in your RNASeq data?

Response: We thank the reviewer for the positive comments. This description was wrong, and what we mean was that PcNCX1 was failed to detected in RNA-seq and the reason may be the bias of mapping algorithm of PcNCX1 reads. Moreover, we detected the expression of PcNCX1 using qRT-PCR and found that the expression level was significantly down-regulated in the sample treated with CA (Figure S1). (line 362-367)

23) Lines 336-343, to elaborate involvement of other pathways in calcium transport upon CA treatment, please add fungal, bacterial or plant related studies.

Response: We agreed the reviewer for the positive comments. We have added two references about Candida albicans and Brassica rapa (line 342-346). The study about the calcium transport pathways upon CA is mostly focused on mammals, but less on fungi, bacteria and plants.

24) Line 337, replace "a" with "an".

Response: We have replaced.

25) Line 349, use "most of the".

Response: We have modified.

26) Line 357, the conclusion is not supported by the data presented. 

Response: We have modified with “suggesting that CA could introduced decreased intracellular Ca2+ level in P. capsici.” (line 368-369)

Reviewer 2 Report

The manuscript is well prepared and the results and discussion are conclusive. The only minor querry is concerned with the citation number 42 (Duffy et al.), which cannot be find in the text.  

Author Response

The manuscript is well prepared and the results and discussion are conclusive. The only minor querry is concerned with the citation number 42 (Duffy et al.), which cannot be find in the text.  

Response: We thank the reviewer for the positive comments. We have modified all the references in the text.

Reviewer 3 Report

The manuscript entitled "The sodium / calcium exchanger PcNcx1-mediated Ca2 + efflux is involved in cinnamaldehyde-induced cell wall defect of Phytophthora capsici" presents the results of a novel series of studies.
The pathogenic fungus P. capsici has a great importance for Solanaceous and Cucurbitaceous hosts, such as: pepper, cucumber, tomato, eggplant, and pumpkin, causing foliar blight, root rot, and fruit rot. Control against it today is mostly based on chemical fungicides.

Cinnamaldehyde (CA) is a natural compound isolated from Cinnamomum cassia. The medicinal properties of CA have been widely identified, but we have quite limited knowledge about its application in plant protection. In this study the the authors present that  CA is significantly inhibited P. capsici in both pepper seedlings and fruits.

The methodology are in state-of-the-art, using RNA analysis and mutant yeast model organisms. Its results and findings are novel, expanding our knowledge of the effects of CA on the detoxifying mechanisms of the pathogenic fungus.

To improve the value of the manuscript, I suggest correcting the following spelling mistakes:

In row 47 cinnamon should be instead of cinnamaon

in row 55 "in vitro" should be written in Italics.

in rows 224, 225 and 281 please correct the scientific name of the pathogen to P. capsici.

In the case of Figure 5., in part A of the histogram, the first line needs to be improved biological instead of biologtical.

Following the implementation of the above corrections, I recommend the publication of the manuscript.

Author Response

The manuscript entitled "The sodium / calcium exchanger PcNcx1-mediated Ca2 + efflux is involved in cinnamaldehyde-induced cell wall defect of Phytophthora capsici" presents the results of a novel series of studies.

The pathogenic fungus P. capsici has a great importance for Solanaceous and Cucurbitaceous hosts, such as: pepper, cucumber, tomato, eggplant, and pumpkin, causing foliar blight, root rot, and fruit rot. Control against it today is mostly based on chemical fungicides.

Cinnamaldehyde (CA) is a natural compound isolated from Cinnamomum cassia. The medicinal properties of CA have been widely identified, but we have quite limited knowledge about its application in plant protection. In this study the the authors present that  CA is significantly inhibited P. capsici in both pepper seedlings and fruits.

The methodology are in state-of-the-art, using RNA analysis and mutant yeast model organisms. Its results and findings are novel, expanding our knowledge of the effects of CA on the detoxifying mechanisms of the pathogenic fungus.

To improve the value of the manuscript, I suggest correcting the following spelling mistakes:

  1. In row 47 cinnamon should be instead of cinnamaon

Response: We have made corrections.

  1. in row 55 "in vitro" should be written in Italics.

Response: We have modified.

  1. in rows 224, 225 and 281 please correct the scientific name of the pathogen to  capsici.

Response: Thanks for the helpful comments. We have corrected all the scientific name to P. capsici.

  1. In the case of Figure 5., in part A of the histogram, the first line needs to be improved biological instead of biologtical.

Response: We have made corrections.

Round 2

Reviewer 1 Report

I thank the authors that improved several parts of the manuscript content. Although the authors put great efforts in this revision of the manuscript such as adding RT-qPCR to show different regulation of PcNCX1 gene, but since they have mentioned they are performing crisper-cas9 to silence respective gene, I highly recommend them to include knockout results to their current manuscript prior to consideration for publication. Additionally, I have added a few comments below for authors' review to reflect in the paper. 

Comment (C)1: The RNASeq is a method to understand the gene expression regulation of an organism. It doesn't have any role in the growth or development of an organism, nor demonstrate mode of action of a compound. Please, revise the added section accordingly. Also, whatever intro about RNASeq should not be at the concluding statement of the introduction (last paragraph).

C5: Since the authors have formerly tried silencing in the P.capsici system, they should include experiments used in the M&M and describe their results, and propose alternative approach (crisper-cas9) in discussion. But since they are developing a crisper cas9 system, I highly recommend to include crisper results into this manuscript. Addition of the later results will enrich the manuscript content and may provide more evidence about the respective gene (BT032616.1) function.

C8: I thank the authors for adding the photos. I should have mentioned in my previous comment that it would be informative if there is one macroscopic and one microscopic views of the mycelial samples in picture 4A. Using only one photo per view per treatment is sufficient.

C9: Please revise this statement accordingly: "After second-strand cDNA synthesis and dual-index adaptor ligation, the paired-end library fragments with 150-200 bp in length .....".

C18: Although, some revision have been made to the discussion, it still lacks the correlation of the current results of PcNCX1 (not the past research) and the literature. For example, discussing about post transcriptional activity in line 389, there is no proof in this study on that. That's why addition of the crisper-cas9 results clarify the role of this gene.

Additional comments:

a) Using RT-qPCR is sound and reasonable due to lack of detection of the gene in RNASeq, but please clarify in the text whether you have used the same aliquot in RT-qPCR that was used for RNASeq library preparation. 

b) Please provide the NCBI accession numbers for the RNASeq reads generated for the treatments.

c) What were the software for trimming the reads, mapping, and gene count? and what was the reference genome used during mapping stage (provide the version and source)? In line 392, authors declared "we can not detected PcNCX1 in the RNA-seq data", I am wondering if it was not differentially expressed (both control and treatment had the similar or close gene counts) or totally the gene count for both treatments was zero.

Author Response

I thank the authors that improved several parts of the manuscript content. Although the authors put great efforts in this revision of the manuscript such as adding RT-qPCR to show different regulation of PcNCX1 gene, but since they have mentioned they are performing crisper-cas9 to silence respective gene, I highly recommend them to include knockout results to their current manuscript prior to consideration for publication. Additionally, I have added a few comments below for authors' review to reflect in the paper. 

Comment (C)1: The RNASeq is a method to understand the gene expression regulation of an organism. It doesn't have any role in the growth or development of an organism, nor demonstrate mode of action of a compound. Please, revise the added section accordingly. Also, whatever intro about RNASeq should not be at the concluding statement of the introduction (last paragraph).

Response: We have changed the description with “On the other hand, it is well known that RNA-seq has been used a routine method for studying the fungal growth, pathogenesis and the genes involved in resistance [18-20].” in line 77-79.

C5: Since the authors have formerly tried silencing in the P.capsici system, they should include experiments used in the M&M and describe their results, and propose alternative approach (crisper-cas9) in discussion. But since they are developing a crisper cas9 system, I highly recommend to include crisper results into this manuscript. Addition of the later results will enrich the manuscript content and may provide more evidence about the respective gene (BT032616.1) function.

Response: Thanks for your suggestion. We tried to use homologous recombination to perform target gene knockout in P. capsici, but failed to get the positive results. So we used yeast mutant as an alternative. We have supplied these information in the section of Materials and Method in revised manuscript. CRISPER-CAS9 is an emerging novel technique for gene knockout. And we just start to try it to manipulate PcNcx1 in P. capsici. Till now, we don’t have any valuable results about it yet. We have proposed CRISPER-CAS9 approach in the section of “Discussion” in revised manuscript (line 374-376), which is a great idea suggested by you.

C8: I thank the authors for adding the photos. I should have mentioned in my previous comment that it would be informative if there is one macroscopic and one microscopic views of the mycelial samples in picture 4A. Using only one photo per view per treatment is sufficient.

Response: We have repeated the experiment and added the macroscopic picture in Figure 4A.

C9: Please revise this statement accordingly: "After second-strand cDNA synthesis and dual-index adaptor ligation, the paired-end library fragments with 150-200 bp in length .....".

Response: We have modified.

C18: Although, some revision have been made to the discussion, it still lacks the correlation of the current results of PcNCX1 (not the past research) and the literature. For example, discussing about post transcriptional activity in line 389, there is no proof in this study on that. That's why addition of the crisper-cas9 results clarify the role of this gene.

Response: Thanks for your suggestion, and we totally agree with you. We have screened the literature, but have found few reports about this gene as well as the involvement of Ca2+ in the antimicrobial activity of CA. Therefore, our current results are pretty novel so that we cannot find enough similar reports to discuss about. There are some reports about the post transcriptional interaction between CA and mammalian Ca2+ channels. Fungal cells are different from mammalian cells, but at least we can propose the possibility of this kind of action. And we have provided the promising approach (e.g. CRISPER-Cas9) to verify this possibility for further study in our future work. Moreover, we deleted the description of “post-transcriptional level”, which was inadequate.

Additional comments:

  1. a) Using RT-qPCR is sound and reasonable due to lack of detection of the gene in RNASeq, but please clarify in the text whether you have used the same aliquot in RT-qPCR that was used for RNASeq library preparation. 

Response: We have repeated the experiment and the aliquot in RT-qPCR was newly processed.

  1. b) Please provide the NCBI accession numbers for the RNASeq reads generated for the treatments.

Response: We thank the reviewer for the positive comments. Unfortunately, we lost the original raw data of RNA-seq due to the damage of Web Server, so we could not upload it to NCBI. However, we can provide the complete sequencing report of this experiment, as well as the analysis results including Assessment of raw data, Mapping Assess, Quantitative Analysis and Structure Analysis. We believe it will help you get acknowledge on the RNA-seq analysis.

  1. c) What were the software for trimming the reads, mapping, and gene count? and what was the reference genome used during mapping stage (provide the version and source)? In line 392, authors declared "we can not detected PcNCX1 in the RNA-seq data", I am wondering if it was not differentially expressed (both control and treatment had the similar or close gene counts) or totally the gene count for both treatments was zero.

Response: FastQC v0.11.3 for trimming the reads, TopHat 2.1.0 for mapping and RSEM v1.2.22 for gene count. The reference genome was obtained from http://genome.jgi-psf.ory/Phyca11/Phyca11.home.html. Due to absence of the raw data of RNA-seq, we suggested that totally the gene count of PcNCX1 for both treatments was zero, because we can not found the data of PcNCX1 (131120) in the expression profile.